# Are We Really Learning the Score Function? Reinterpreting Diffusion Models Through Wasserstein Gradient Flow Matching

**An B. Vuong**[*]                                                                           *vuonga2@oregonstate.edu*
*Department of Electrical Engineering & Computer Science*
*Oregon State University*

**Michael T. McCann**                                                                        *mccann@lanl.gov*
*Los Alamos National Laboratory*

**Javier E. Santos**                                                                         *jesantos@lanl.gov*
*Los Alamos National Laboratory*

**Yen Ting Lin**                                                                             *yentingl@lanl.gov*
*Los Alamos National Laboratory*

**Reviewed on OpenReview:** *https://openreview.net/forum?id=CzyJqXQRhJ*

## Abstract

Diffusion models are commonly interpreted as learning the *score function*, i.e., the gradient of the log-density of noisy data. However, this learning target is a conservative vector field (i.e., a vector field that is the gradient of some function), a property not enforced by neural network architectures used in practice. We show numerically that trained diffusion networks violate both the integral and differential constraints that conservative vector fields must satisfy, indicating that the learned vector fields are not score functions of any density. Despite this, the models perform remarkably well as generative mechanisms. To explain this paradox, we propose a new theoretical perspective: diffusion training is better understood as *flow matching* to the velocity field of a Wasserstein Gradient Flow (WGF), rather than as score learning for a reverse-time stochastic differential equation. Under this view, the "probability flow" arises naturally from the WGF framework, eliminating the need to invoke reverse-time SDE theory and clarifying why generative sampling remains successful, even when the neural vector field is not a true score. We further show that non-conservative errors from neural approximation do not necessarily harm density transport. Our results advocate adopting the WGF perspective as a principled, elegant, and theoretically grounded framework for understanding diffusion generative models.

## 1 Introduction

Diffusion models are typically described as follows: Given $D$-dimensional samples $x \in \mathbb{R}^D$ drawn from a data distribution $\mu_0$, one defines a forward Itô process that gradually corrupts $x$ into noise. Throughout this paper, we use the continuous-time Ornstein–Uhlenbeck (OU) process for concreteness:[1]

$$\mathrm{d}X_t = -X_t\,\mathrm{d}t + \sqrt{2}\,\mathrm{d}W_t, \qquad X_0 = x \sim \mu_0, \tag{1}$$

---

[*]Work partially done during the Applied Machine Learning Summer School at LANL.

[1]Santos & Lin (2023) established the equivalence of the OU process with the discrete-time Denoising Diffusion Probabilistic Model Ho et al. (2020) and the score-based formulation (Song et al., 2021). This setup is often called "variance-preserving" (VP), though this term is misleading: for each sample, the variance is not constant over time (which, in most scientific contexts, is the definition of "preserving"), but grows as $\sqrt{1 - e^{-2t}}$. Our analysis extends naturally to the standard Brownian motion process $\mathrm{d}X_t = \mathrm{d}W_t$, termed "variance-exploding" (VE) by Song et al. (2021).

where each component of $W_t$ is a standard $D$-dimensional Wiener process. The process Eq. (1) converges to the limiting distribution $\mu_\infty$ as $t \to \infty$, which is an isotropic Gaussian in $\mathbb{R}^D$. Due to the diagonal form of the drift and diffusion terms, each component of $X_t$ follows the canonical one-dimensional OU process.

Equivalently, the forward dynamics can be described in terms of densities. The transition kernel[2] $\rho(\xi, t|\zeta, s)$ satisfies the Fokker–Planck Equation (FPE):

$$\partial_t \rho(\xi, t|\zeta, s) = \nabla_\xi [\xi \, \rho(\xi, t|\zeta, s)] + \nabla_\xi^2 \rho(\xi, t|\zeta, s), \tag{2}$$

with the initial condition $\rho(\xi, 0) = \delta(\xi - x)$ for each of the drawn samples $x \sim \mu_0$, where $\delta(\cdot)$ denotes the Dirac delta distribution.

The modern understanding of diffusion models is grounded in Anderson's theory (Anderson, 1982), which guarantees the existence of a reverse-time Itô process that transforms samples from the simple, $D$-dimensional isotropic Gaussian, distribution $\mu_\infty$ back into data-like samples as $t : \infty \to 0$:

$$dX_\tau = [X_\tau + 2s\,(X_\tau, -\tau)]\,d\tau + \sqrt{2}\,dW_\tau, \quad X_{-\infty} \sim \mu_\infty. \tag{3}$$

Here, we define $\tau := -t$, $\tau : -\infty \to 0$, $\rho(x, t)$ denotes the forward density with initial distribution $\mu_0$, $s(\xi, t) := \nabla_\xi \log \rho(\xi, t) \in \mathbb{R}^D$ is the score function of the corrupted (forward) distribution given the initial distribution $\mu_0$, and $dW_\tau$ is again a $D$-dimensional Wiener process. The central training objective of diffusion models is thus framed as *learning the score function* $s(x, t)$ (Song et al., 2021). In practice, a neural network $\mathbb{R}^D \times \mathbb{R} \to \mathbb{R}^D$ is used to approximate $s(x, t)$, which is then plugged into Eq. (3) during sampling.

A key point is that the score function has a distinct mathematical structure: it is a conservative field, the gradient of a scalar field $\log \rho$. Neural networks used in practice are not constrained to produce conservative vector fields and, therefore, do not necessarily preserve this structure. This raises the central question of this study:

> Does a trained neural network actually learn a valid *score function*, or merely a useful vector field for generative sampling?

## 2 Literature Review

Our work attempts to connect the theory of Diffusion Models (DMs) and Wasserstein Gradient Flows (WGFs). Diffusion-based generative modeling originated from the observation that adding noise to data and then removing it can reveal its underlying structure, a principle first formalized in the denoising autoencoder by Bengio et al. (2013), which viewed training as learning to invert a stochastic corruption process. This idea was later generalized by Sohl-Dickstein et al., establishing a forward–reverse Markov chain formulation that defines data generation as a learned inversion of a diffusion process. The seminal Denoising Diffusion Probabilistic Model (DDPM) proposed by Ho et al. (2020) greatly simplified and stabilized this framework by adopting fixed Gaussian noise schedules and a mean-squared "denoising" objective, hinting at a connection between diffusion model training and denoising score matching (Hyvärinen, 2005; Vincent, 2011). Subsequently, Song et al. (2021) generalized the approach in continuous time using Itô stochastic differential equations (SDEs), showing that diffusion models can be viewed as learning the score function $s(\xi, t)$ of noisy data distributions and enabling sampling via reverse-time SDEs, established by Anderson (1982). Ghimire et al. (2023) investigated the Riemannian structure underlying score-based generative models, offering valuable insights into their manifold geometry. However, their goal was the geometric analysis of existing models, not a reinterpretation of the algorithm through our proposed WGF flow matching. Together, these developments established the standard score-based generative modeling pipeline of modern DMs.

Recent research has extended the diffusion paradigm beyond Gaussian noise and continuous states. Campbell et al. (2022) formulated a continuous-time limit for discrete-state denoising diffusion, while Santos et al.

---

[2]Since the OU process decomposes into $D$ independent one-dimensional processes, the density factorizes across coordinates: $\rho(\xi, t \mid \zeta, s) = \prod_{i=1}^{D} \rho_i(\xi_i, t \mid \zeta_i, s)$

(2023) developed an exact theoretical framework for arbitrary discrete-state Markov processes without variational approximations, deriving the discrete-time analog of Anderson's reverse-time SDE. We remark that this study does not extend to these generalized DMs which leverage non-Gaussian noise.

Wasserstein Gradient Flow (WGF) originates from the theory of optimal transport (OT), but it has become increasingly central to the theoretical understanding of modern generative models. Classic monographs such as Ambrosio et al. (2008) and Figalli & Glaudo (2023) provide comprehensive foundations for this framework. The connection between WGF and diffusion models lies in the forward evolution of the probability density $\rho(x,t)$, governed by the Fokker–Planck equation (FPE (2)). In their seminal work, Jordan et al. (1998) demonstrated that an implicit Euler discretization of the FPE can be reinterpreted as a variational problem: each timestep corresponds to minimizing a free-energy functional composed of a potential energy term $V(x)$ and a negative Shannon entropy of the evolving distribution, plus a Wasserstein-2 distance term between successive densities (scaled by the time step). The entropy and the Wasserstein-2 distance can both be viewed as regularizers of the optimization problem for the potential energy. Moreover, the sum of the potential energy and the negative entropy is equivalent to the Kullback—Leibler divergence $\text{KL}(\rho_t\|\pi)$ from the evolving probability density $\rho_t$ to the Gibbs measure $\pi(x) \propto \exp(-V(x))$. This insight—known as the JKO scheme—established that the FPE can be viewed as a gradient flow in the space of probability measures. Building on this foundation, Otto (2001) introduced a formal Riemannian calculus on the manifold of probability distributions, showing that the FPE describes the steepest descent of free energy under Wasserstein geometry. This formulation, now widely known as Otto calculus, provides a rigorous mathematical language for describing the evolution of probability densities as particles slide down an energy landscape shaped by optimal transport. Furthermore, Otto introduced the generalized Liouville equation[3] (GLE, Gerlich (1973)) and establishes the link between macroscopic density evolution and microscopic transport dynamics. Together, the JKO scheme and Otto's formulation provide the foundation for WGF, unifying PDE evolution, entropy maximization, and optimal transport.

More recently, Wasserstein-gradient concepts have gained renewed attention across machine learning, particularly in sampling, variational inference, and generative modeling. Early work by Wibisono (2018) framed Langevin dynamics as optimization in the space of probability measures, highlighting the close relationship between stochastic sampling and gradient flow in Wasserstein geometry. Subsequent studies extended this principle to practical optimization and inference algorithms in high dimensions, such as the Wasserstein proximal gradient (Salim et al., 2020) and variational WGF formulations (Lambert et al., 2022; Fan et al., 2022). These methods treat WGF primarily as a computational framework for density evolution or inference, rather than as a reinterpretation of diffusion-based generative learning. Other lines of work have explored large-scale and neural-network-based implementations of Wasserstein flows (Mokrov et al., 2021; Alvarez-Melis et al., 2022), and recent efforts have proposed scalable or unbalanced formulations for generative modeling (Choi et al., 2024). Despite these advances, the majority of these studies focus on constructing or optimizing explicit Wasserstein flows rather than reinterpreting existing diffusion models within the WGF framework.

## 3   Numerical experiments

We first perform numerical experiments to verify the central question we have in score-based generative modeling: *Are we learning the score function?*

Due to the definition of the score function, $s(x,t) := \nabla_x \log \rho(x,t)$, the fundamental theorem of calculus (or generalized Stokes' theorem in high dimensions) states that the line integral of the score function along a closed path in the state space must be equal to zero:

$$\oint \vec{s}(x,t) \cdot \mathrm{d}\vec{x} = 0. \tag{4}$$

---

[3]We distinguish GLE from the "continuity equation", a term more broadly used in the physics literature to describe conservation of various quantities, such as mass, heat, etc. GLE is a special case of continuity equations that describes the evolution of normalized probability density functions.

We refer to Eq. (4) as the *integral constraint.* The second constraint, also following directly from the definition of the score function, states:

$$\frac{\partial}{\partial x_j} s_i(x,t) = \frac{\partial}{\partial x_i} s_j(x,t), \quad \text{for any pair } (i,j) \in \{1 \ldots D\}^2 \, . \tag{5}$$

We refer to Eq. (5) as the *differential constraint.* Our goal is to numerically investigate whether either of the constraints is met in trained diffusion models.

### 3.1 Models and datasets

To present a minimal working example, we trained an MNIST diffusion model using a lightweight U-Net implementation. The model is composed of ShuffleNet-style residual bottlenecks and depthwise convolutions. The time indices are embedded, passed through an MLP, and added to the feature maps in each block. It employs simple encoder–decoder blocks with downsampling, upsampling, and skip connections, keeping the model lightweight (around 4 MB)[4]. We used the cosine schedule (Nichol & Dhariwal, 2021) and a total discrete time index $T = 1000$, which corresponds to observing time-homogeneous OU process (1) at discrete times (Santos & Lin, 2023):

$$t_k = -\frac{1}{2} \log \frac{f(k)}{f(0)}, \ \ f(k) := \cos\left(\frac{k/T + 0.008}{1 + 0.008}\frac{\pi}{2}\right). \tag{6}$$

We also performed the same test with latent diffusion, using a VAE with an $8\times8$ latent space[5]. The diffusion process employs the same network as before but operates in the latent space of the VAE.

In addition to the MNIST dataset, we perform experiments on several other datasets and neural architectures, including publicly available state-of-the-art Diffusion Models:

- CIFAR-10, using a latent DDPM.

- CelebA-HQ-256, using the pretrained variance-exploding model from Song et al. (2021).

- Neal's funnel distribution (Neal, 2003), using the same lightweight neural architecture as in the MNIST experiment.

Our motivation for carrying out a comprehensive numerical analysis is to demonstrate the validity of our claims on popular high-dimensional datasets in simple settings where the data distribution is analytical, using commonly used neural architectures. As we observed very similar behavior, we present the results of the MNIST dataset in Fig. 2 in the main text and the rest in the Appendices 6.2 and 6.3.

### 3.2 Integral constraints

To numerically evaluate the integral constraint (Eq. (4)), we introduce three mechanisms for generating closed paths over which the integral is computed:

- **Brownian path.** Starting from a corrupted sample $x_t \in \mathbb{R}^D$ generated by the forward diffusion, we perform a random walk on $\mathbb{R}^D$ using a Brownian bridge, which generates a path in $\mathbb{R}^D$ starting and ending at $x_t$. The path of a Brownian bridge is $X_u^{\text{BB}} = W_u - uW_U/U$ with a fictitious time $u \in [0, U)$. We choose $U = 9$, uniformly sample 1,000 discrete time steps in between, and add the resulting path to a forward sample $x_t$, i.e., $y_{u;t} = x_t + X_u^{\text{BB}}$. This method does not guarantee that the path stays close to the the typical region induced by the forward process, as illustrated in Fig. 1 (a). We include this path as a way to study the behavior of out-of-distribution samples.

---

[4]The neural network implementation can be found at `https://github.com/bot66/MNISTDiffusion`.

[5]Implementation based on `https://github.com/sksq96/pytorch-vae/blob/master/vae.py`.

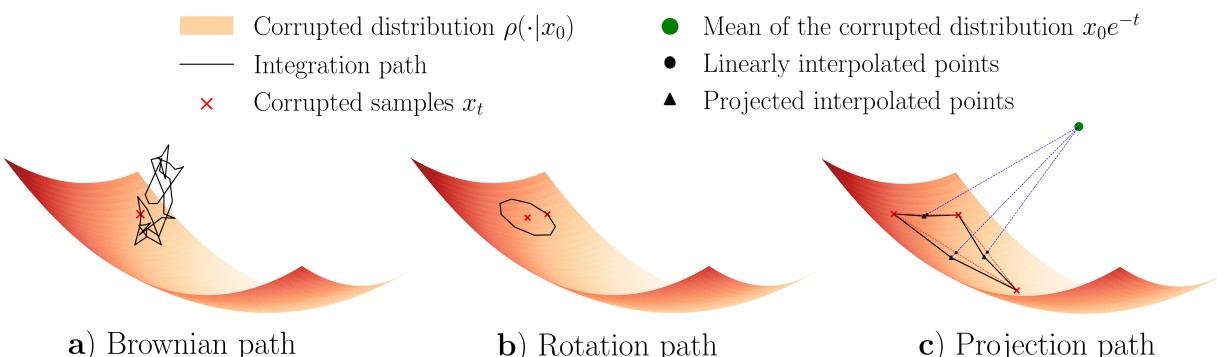

Figure 1: Mechanisms for assessing integral constraints. Illustration of the three mechanisms we used to construct closed paths for evaluating integral constraints within the high-density regions of the data distribution.

- **Rotation path.** Following the typical application of image corruption process, the corrupted sample $x_t = x_0 e^{-t} + \sqrt{1 - e^{-2t}}\varepsilon$, where $\varepsilon \sim \mathcal{N}(0, I)$. We randomly pair each of the $D$ components of $\varepsilon$, so $(\varepsilon_i, \varepsilon_j)$ forms a two-dimensional vector. Then, we rotate each of the $D/2$ two-dimensional vectors with respect to the origin, i.e., $\varepsilon_i'(u) = \cos(2\pi u)\varepsilon_i + \sin(2\pi u)\varepsilon_j$ and $\varepsilon_j'(2\pi u) = -\sin(2\pi u)\varepsilon_i + \cos(2\pi u)\varepsilon_j$. Note that we rotate all $D/2$ pairs with the same "angular velocity". The resulting vector is used to generate a closed loop in the $x$-space, i.e., $y_{u;t} = x_0 e^{-t} + \sqrt{1 - e^{-2t}}\varepsilon'(u)$, $u : 0 \to 1$. With this construct, the probability density of noise realization $\varepsilon'(u)$ is identical to that of the original noise realization $\varepsilon$, ensuring the closed path in the $x$-space sits in the region where most of the probability mass is.

- **Projection path.** We first generate multiple corrupted samples $x_t$ from the same initial $x_0$, then find a way to connect these points such that the connections lie in the typical set of corrupted distribution. To achieve this, we propose a simple mechanism: to connect two corrupted samples $x_t$ and $x_t'$, we first generate points that linearly interpolate between the two samples, and then project the interpolated points back to the corrupted distribution. Since Gaussian diffusion in high-dimensional space induces the structure of a thin shell around the clean samples, the projection can be carried out by projecting the samples radially back to the shell in $\mathbb{R}^D$, whose radius is estimated through Monte Carlo sampling (which we also know would be $\approx \sqrt{D}$ from asymptotic analysis). An illustrative schematic diagram is provided in Fig. 1 (c).

Figures 2 (a) and (b) show the results of evaluating the integral constraint using these three methods of generating closed paths on MNIST. Figure 3.(a-b) reports the same statistics for the funnel and CelebA-HQ-256. Summary statistics of these distributions are provided in Fig. 5 & Fig. 12 in the Appendix.

Clearly, the integral condition is not satisfied in the trained neural network. One may argue whether the magnitude matters for the reverse-time dynamics. To answer this, we notice that the score-induced drift $2s(x, t)$ is added to a linear term $x(t)$ in Eq. (3); this provides us a non-dimensional quantity:

$$\frac{2 \oint \vec{s}(\vec{y}, t) \cdot \mathrm{d}\vec{y}}{\oint |\vec{y}| \, |\mathrm{d}\vec{y}|}, \tag{7}$$

where $\vec{y}$ is a dummy vector looping over the generated path. Results of this quantity are presented in Figs. 6 and 7 in the Appendix, showing a significant deviation from 0.

### 3.3 Differential constraints

Because computing the full Jacobian matrix is computationally intensive, we instead randomly sample 64 components of the predicted score $s(x, t)$ and 64 components of the corrupted samples $x_t$ to compute a $64 \times 64$ sub-Jacobian matrix using `torch.func.jvp` functionality. The statistics for MNIST were collected from 256 samples for each time step and are presented in Fig. 2 (c) and (d), both showing non-zero contributions.

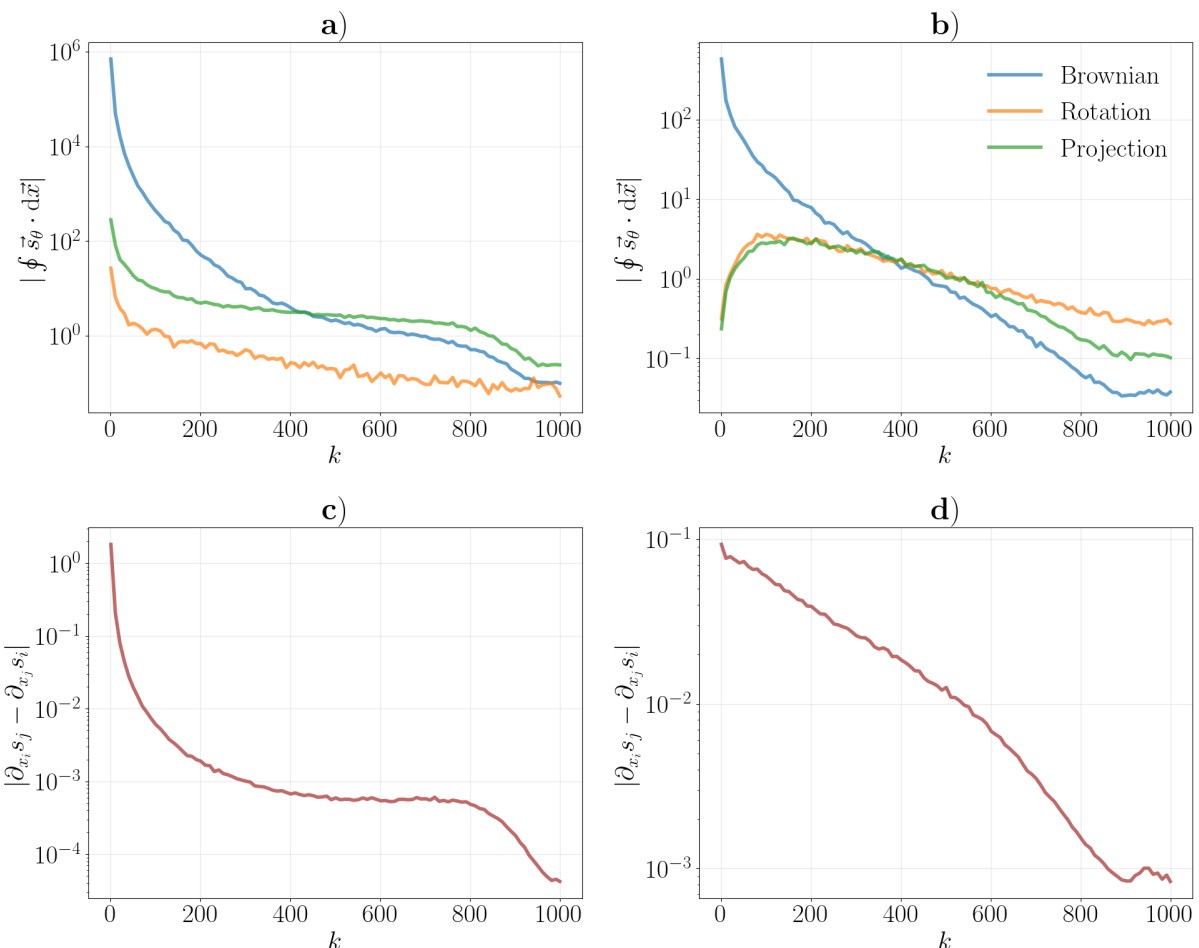

Figure 2: (**MNIST** - **Left**: Pixel space, **Right**: Latent space) Results of integral and differential constraints, as functions of discrete time index $k$: **a)** shows the absolute value of the integral condition $\oint \vec{s}_\theta \cdot \mathrm{d}\vec{x}$; **b)** presents the same quantity but for the latent dynamics; **c)** reports the differential condition $|\partial_{x_i} s_j - \partial_{x_j} s_i|$ in normal diffusion; **d)** shows the corresponding differential condition in latent diffusion.

For CelebA-HQ-256, given the scale of the input and the presence of custom layers in the implementation from Song et al. (2021), running `torch.func.jvp` was computationally infeasible. Consequently, we utilized finite-difference methods to estimate the sub-Jacobian components for this setting. Results are presented in Fig. 3 (d).

The observed behavior is strikingly consistent across all experiments and configurations. From analytically tractable distributions such as the funnel distribution to increasingly complex real datasets including MNIST, CIFAR-10, and CelebA, and across a wide range of neural architectures, our lightweight U-Net, the latent-diffusion VAE, the DDPM baseline, and the publicly released variance-exploding (VE) model in (Song et al., 2021) we consistently observe the same violations of both integral and differential score-function constraints. This consistency persists under both variance-preserving (VP) and variance-exploding (VE) diffusion setups. Together, these results rule out the possibility that our finding—that trained neural networks in diffusion models do not learn truly conservative score fields—is an artifact of a particular dataset, model capacity, or diffusion configuration. Instead, it points to a robust and universal characteristic of current diffusion-model training practices.

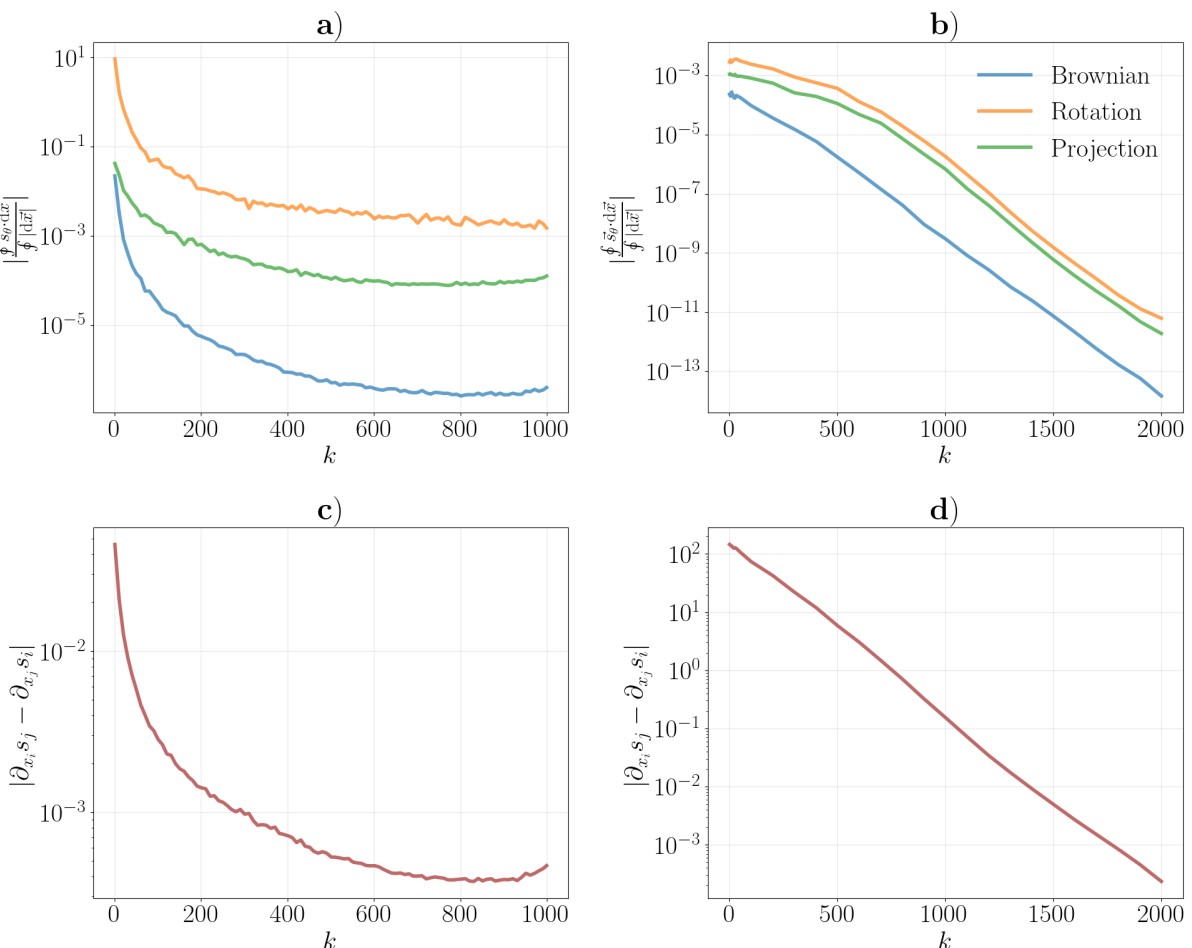

Figure 3: (**Left**: Funnel, **Right**: CelebA-HQ-256) Results of integral and differential constraints, as functions of discrete time index $k$: **a)** shows the absolute value of the integral condition $\oint \vec{s}_\theta \cdot \mathrm{d}\vec{x}$ normalized by the path length on data samples drawn from funnel distribution; **b)** presents the same quantity but for the CelebA-HQ-256 dataset; **c)** and **d)** show the differential condition $|\partial_{x_i} s_j - \partial_{x_j} s_i|$ for these datasets respectively. We report the normalized integral here because the variance is not controlled (variance-exploding) in this setting, so normalizing by path length makes the comparison fairer between each methods.

## 4 Connecting Diffusion Models with Wasserstein Gradient Flow Theory

The numerical evidence clearly suggests that *the trained neural network* **does not** *learn the score function*, which is a conservative field. However, the trained network can definitely perform the generative task. The observation raises an interesting question: what is the trained neural network actually learning in order to perform the generative task?

We propose a bold hypothesis, leveraging the WGF theory, to understand what happens in the diffusion modeling. Our assertion is:

> Existing diffusion modeling is better understood as modeling a normalizing flow (Chen et al., 2019), through performing flow matching (Lipman et al., 2022) to the WGF velocity (Eq. (13)), rather than learning the reverse stochastic differential equation established by Anderson (1982) and popularized by Song et al. (2021).

We emphasize that this interpretation does not invalidate the reverse-time SDE framework, which remains *mathematically correct when the true score function is available.* Rather, the WGF formulation provides a complementary and unifying perspective that explains why diffusion models remain empirically robust, even when the learned vector field deviates from a perfectly conservative score. In this view, standard diffusion training can be regarded as performing approximate flow matching to the WGF velocity field, ensuring correct marginal density transport despite imperfections in the neural approximation. This complementary interpretation highlights the consistency between the WGF and SDE descriptions: when the score is exact, both yield identical dynamics; when it is approximate, the WGF view naturally accounts for the observed robustness of diffusion-based generative models.

We now shift our focus to establishing a direct theoretical correspondence between the learned vector field in diffusion training and the velocity field of a Wasserstein Gradient Flow. This reinterpretation explains why modern diffusion networks remain empirically successful even when the learned field is not guaranteed to be conservative, and it situates diffusion modeling within a principled continuum of variational and geometric formalisms. One powerful result of WGF theory is:

> While the sample paths of the diffusion process that FPE describes are fundamentally *stochastic*, the marginal distribution[6] of the paths at a specific time, $\rho(\cdot, t)$, is identical to the marginal distribution of the trajectories driven by a deterministic WGF.

In the context of diffusion modeling with Ornstein–Uhlenbeck forward process, let us consider setting the energy functional as the sum of a quadratic potential and the negative Shannon entropy

$$E\left\{\rho\left(\cdot,t\right)\right\} := \int \frac{x^2}{2}\rho\left(x,t\right)\mathrm{d}x + \int \rho\left(x,t\right)\log\rho\left(x,t\right)\,\mathrm{d}x. \tag{8}$$

Here, the first term accounts for the drift/advection and the second for the diffusion in the FPE (2). The goal is to identify the *steepest descent* direction that minimizes the energy the most in the space of probability density functions induced by a deterministic velocity field $v(x,t)$. Applying $\mathrm{d}/\mathrm{d}t$ to the energy functional:

$$\frac{\mathrm{d}}{\mathrm{d}t}E\left\{\rho\left(\cdot,t\right)\right\} = \int \frac{\delta E\left\{\rho\left(\cdot,t\right)\right\}}{\delta\rho(x,t)}\frac{\partial\rho\left(x,t\right)}{\partial t}\mathrm{d}x, \tag{9}$$

where the functional variation of $E$ with respect to the density function $\rho$ can be explicitly computed:

$$\frac{\delta E\left\{\rho\left(\cdot,t\right)\right\}}{\delta\rho(x,t)} := \frac{1}{\delta\rho(x,t)}\left[\int \frac{x^2}{2}\delta\rho\left(x,t\right) + \left(\rho+\delta\rho\right)\log\left(\rho+\delta\rho\right)\,\mathrm{d}x - \int \rho\left(x,t\right)\log\rho\left(x,t\right)\,\mathrm{d}x\right]$$

$$\sim \frac{1}{\delta\rho(x,t)}\int \left[\frac{x^2}{2} + \log\rho(x,t) + 1\right]\delta\rho\left(x,t\right)\,\mathrm{d}x = \frac{x^2}{2} + \log\rho(x,t). \tag{10}$$

In the last two equations, we neglected higher-order $\mathcal{O}(\delta\rho(x,t))$ terms (using the asymptotic symbol $\sim$) and applied the normalization condition that the functional perturbation $\int \delta\rho(x,t)\,\mathrm{d}x = 0$ because $\int \rho(x,t)\,\mathrm{d}x = 1 = \int \left(\rho+\delta\rho\right)(x,t)\,\mathrm{d}x$. Next, inserting GLE (see footnote 3):

$$\partial_t\rho\left(x,t\right) = -\nabla_x \cdot \left[v(x,t)\rho(x,t)\right], \tag{11}$$

and the functional variation Eq. (10) into Eq. (9) leads to

$$\frac{\mathrm{d}}{\mathrm{d}t}E\left\{\rho\left(\cdot,t\right)\right\} = -\int \left[\frac{x^2}{2} + \log\rho\left(x,t\right)\right]\nabla_x \cdot \left[v(x,t)\rho(x,t)\right]\,\mathrm{d}x$$

$$= \int v(x,t) \cdot \left[x + \nabla_x\log\rho\left(x,t\right)\right]\rho\left(x,t\right)\mathrm{d}x, \tag{12}$$

where we used integration by parts and assumed vanishing boundary terms. The above equation can be interpreted as an inner product of the functions $v(\cdot,t)$ and $\nabla_x\log\rho(\cdot,t)$ under the measure $\rho(\cdot,t)$. Clearly,

---

[6] $\rho(\cdot,t)$ is referred to as the marginal distribution because it is only the distribution of $X_t$ at time $t$. It is a marginal distribution of the joint distribution specified the stochastic process, $\rho\left(x_{t_1},\ldots x_{t_N}\right)$.

the velocity field that corresponds to the steepest descent of the energy functional should align with the opposite direction of $\nabla_x \log(\cdot, t)$ (up to a global multiplicative constant):

$$v_{\text{WGF}}(x, t) := -x - \nabla_x \log \rho(x, t) = -x - s(x, t). \tag{13}$$

The probability distribution of the resulting flow system with the above velocity field evolves under the GLE:

$$\frac{\partial}{\partial t} \rho(x, t) = -\nabla_x \cdot [v_{\text{WGF}}(x, t) \rho(x, t)] = \nabla_x \left[ (x + \nabla_x \log \rho(x, t)) \, \rho(x, t) \right], \tag{14}$$

which is exactly the FPE (2) describing the OU.

We remark that Song et al. (2021) rediscovered the WGF velocity field (Eq. (13)) through manipulating the reverse-time FPE and noticing $\nabla_x \rho(x, t) = \rho(x, t) \nabla_x \log \rho(x, t)$. They did not recognize the fundamental WGF structure of the forward process, as we illustrated above. They used the term "probability flow", without referencing the JKO scheme, Otto calculus, and WGF. We believe it is beneficial to point out the origin of this theoretical framework, given its deeper connection to OT and the variational nature of the diffusion process. We also remark that in a recent pre-print by Ghimire et al. (2023), the equivalence between diffusion models and WGF was also reported.

Now that the structure of the forward WGF is established, we turn our attention to how we can use Flow Matching (Lipman et al., 2022) to learn the WGF. Contrary to typical flow-based models (Chen et al., 2019), which learn the velocity field by maximizing the end-to-end likelihood, the flow-matching method (Lipman et al., 2022) matches the neural velocity field to a target velocity field at all times in a continuous time domain, connecting the initial and final distributions. The target velocity field is often analytically derived for a prescribed transport from the data distribution to an easy-to-sample distribution (often an isotropic Gaussian distribution in high dimension), and evaluated on sampled training data. Here, we use the WGF induced by the energy functional (Eq. (8)) as the prescribed transport, and match its velocity field (Eq. (13)). More precisely, we only match the flow induced by the entropic term in Eq. (8).

There are several advantages to understanding the diffusion model as the flow-matching WGF. First, the "probability flow" is trivially included in the WGF framework. Secondly, we can formally bypass the necessity to invoke the reverse-time Itô process, which can be confusing and counterintuitive. As will be seen below, within the WGF and Otto calculus framework, the deterministic probability flow ODE arises trivially: it is just the time-reversal of the forward WGF velocity Eq. (13). Consequently, the theory of learning and inference is elegantly simple, without the need to explicitly route through Anderson's reverse-time SDE. Finally, flow-matching WGF naturally explains why the trained neural flow, which fails to obey the differential and integral score conditions, can still perform well in generative modeling.

Let us now introduce a self-consistent narrative of a flow-matching problem:

1. **Training.** Our goal is to learn the WGF Eq. (13) through flow-matching. We minimize the $L^2$ error between the neural velocity field $v_\theta$ and the entropy-induced velocity field $s(\cdot, t)$ in Eq. (13):

$$\min_\theta \mathbb{E}_{t \sim \text{Unif}(0, T)} \mathbb{E}_{x \sim \rho(\cdot, t)} \| v_\theta(\cdot, t) - s(\cdot, t) \|_2^2. \tag{15}$$

Here, we remark that the target time for performing the "matching" is drawn from *a priori* selected distribution (here, $\text{Unif}(0, T)$) between the initial (0) and final ($T$) times. In practice, we would choose $T \gg 1$ such that the distribution $\rho(\cdot, T)$ can be reasonably approximated by an isotropic Gaussian distribution, which is the limiting distribution, formally correct only at $T \to \infty$. For discrete-time algorithms such as in DDPM, discrete times $\{t_k\}_{k=1}^T$ are often selected uniformly, but discrete times $t_k$'s are often non-uniformly distributed in the time domain due to various noise scheduling functions (Santos & Lin, 2023).

(a) **Data generation.** Samples to perform Monte Carlo approximation of the above $L^2$-norm will be drawn from the distribution at time $t$, induced by the energy function (Eq. (8)). Instead of using the WGF in the forward dynamics, which requires knowing $s(x, t)$—the object we attempt to learn in high

dimension—we use the equivalent OU process (Eq. (1)) to generate samples. Due to the stochastic nature of the OU process, we can formally generate infinitely many samples at any arbitrary time for parametrizing $v_\theta$. Also, due to the existence of an analytical expression of the OU process (Santos & Lin, 2023), samples can be generated efficiently.

(b) **Parametrizing $v_\theta$ in practice.** In general, we do not have the training labels $s(x, t) := \nabla_x \log \rho(x, t)$ for the randomly generated samples $x$ at time $t$. The difficulty can be solved by *denoising score-matching* proposed by Vincent (2011) which transforms learning $v_\theta$ using the instantaneous distribution $\rho(x_t)$ to another learning problem using the joint (*two-time*) distribution $\rho(x_t, x_0)$,

$$\min_\theta \mathbb{E}_{t \sim \mathrm{Unif}(0,T)} \mathbb{E}_{x_t, x_0 \sim \rho(x_t, x_0)} \|v_\theta(\cdot, t) - s(x_t|x_0)\|_2^2, \tag{16}$$

whereas the "labels" for training becomes the conditional score function $s(x_t|x_0) := \nabla_x \log \rho(x_t|x_0)$, which is analytical for this OU process, evaluated at a randomly sampled time $t$. We remark that the choice of $L^2$ cost function is necessary to ensure that the minimizer of the problem Eq. (16) is the same as that of the problem Eq. (15).

2. **Sampling/Inference.** To perform the generative task, terminal samples drawn from the isotropic Gaussian are transported from $t \to \infty$ to $t = 0$ by integrating the ordinary differential equation backward in time. That is, $\mathrm{d}x(\tau)/\mathrm{d}\tau = -v_{\mathrm{WGF}}(x(\tau)) = x(\tau) + \mathrm{NN}(x(\tau), -\tau)$, where $\tau \equiv -t$, so signs flip relative to forward time. $x(\infty) \sim \mathcal{N}(0, I)$ and $\tau : -\infty \to 0$. The corresponding GLE (Gerlich, 1973) is:

$$\frac{\partial}{\partial \tau} \rho(x, \tau) = -\nabla_x [(x + v_{\theta^*}(x, -\tau)) \rho(x, \tau)], \tag{17}$$

where $\theta^*$ stands for the trained neural weights.

Operationally, the above descriptions are identical to applying the "score-matching" for training and performing "probability flow" for inference (Song et al., 2021). However, due to the deterministic nature of the WGF, we would not need to invoke the reverse-time stochastic process (Anderson, 1982) in the theory. The simplicity is the primary benefit of recognizing the existing approach as a Wasserstein gradient flow-Matching problem.

By framing the learning as a flow-matching problem, it is most natural to weight each time equally, which is the *de facto* training procedure for both discrete-time (Ho et al., 2020) and continuous-time (Song et al., 2021) diffusion models. The procedure would seem *ad hoc* if one aims to parameterize a neural network for learning the reverse-time diffusion process by a more theoretically grounded log-likelihood (more precisely, the bound of which) maximization as shown by Sohl-Dickstein et al. As Ho et al. (2020) pointed out, the log-likelihood approach involved weights which are not uniform in time; by removing such non-uniform weights, DDPM achieved a better performance by effectively solving a flow-matching problem.

Next, assuming that we learn the WGF perfectly, we can treat the reverse-time WGF as a dynamical system:

$$\frac{\mathrm{d}}{\mathrm{d}\tau} x(\tau) = x(\tau) + \mathrm{NN}(x(\tau), -\tau) = x(\tau) + \nabla_x \log(x(\tau), -\tau). \tag{18}$$

This system is identical to a Wasserstein Gradient Flow with the energy functional,

$$E\{\rho(\cdot, \tau)\} = -\int \frac{x^2}{2} \rho(x, \tau) \, \mathrm{d}x - \int \rho(x, \tau) \log \rho(x, \tau) \, \mathrm{d}x$$

$$= \underbrace{-\int \frac{x^2}{2} \rho(x, \tau) \, \mathrm{d}x - 2 \int \rho(x, \tau) \log \rho(x, \tau) \, \mathrm{d}x}_{\text{Reverse-time drift}} + \underbrace{\int \rho(x, \tau) \log \rho(x, \tau) \, \mathrm{d}x}_{\text{Reverse-time diffusion}} \tag{19}$$

which is equivalent to the reverse-time Itô process (Eq. (3)). This suggests that we would not need to invoke Anderson (1982)'s seminal proof of the existence of the reverse diffusion for generative task. This justifies the second advantage of the WGF framework. We remark, however, that to rigorously establish the equivalence of the forward and reverse *path measures*, Anderson's theory remains necessary. Nevertheless,

because generative diffusion models only require consistency at the level of marginal densities, it is not necessary to invoke path measures in practice. We emphasize that our results concern density transport (marginals). We do not make claims about sample-path equivalence, which requires Anderson's reverse-time construction. However, the corresponding reverse-time Itô process can not only can be used as a stochastic process for sampling, but also coincidentally the reverse-time process established by Anderson (1982).

Finally, as suggested by our numerical analysis, the neural network is *not* learning a gradient of a scalar potential, i.e. $\text{NN}(x,t) \neq s(x,t)$ for all $t$, both globally (because it violates the integral conditions) or locally (because it violates the differential conditions.) It is thus puzzling and challenging to analyze how the violations affect the reverse-time diffusion, and consequently, the quality of the generated samples. The flow representation can provide some insight here. Suppose we use the trained, yet imperfect neural velocity field $\text{NN}(x,t) \approx \nabla_x \log \rho(x,t)$. Denote the error by $e(x,t) := s(x,t) - \text{NN}(x,t)$. Then, the GLE governing the distribution driven by the neural velocity field is

$$
\begin{aligned}
\frac{\partial}{\partial \tau} \rho(x, \tau) = & -\frac{\partial}{\partial x} \left[ (x + \text{NN}(x, -\tau)) \rho(x, -\tau) \right] \\
= & -\frac{\partial}{\partial x} \left[ (x + s(x, -\tau)) \rho(x, -\tau) \right] + \frac{\partial}{\partial x} \left[ e(x, -\tau) \rho(x, -\tau) \right] \\
= & -\frac{\partial}{\partial x} \left[ (x + s(x, -\tau)) \rho(x, -\tau) \right] \\
& + \left[ \nabla_x \cdot e(x, -\tau) + s^T(x, -\tau) \cdot e(x, -\tau) \right] \rho(x, -\tau).
\end{aligned}
\tag{20}
$$

Immediately, we can identify a condition that if the error field $e(x,t)$ satisfies

$$
0 = \nabla_x \cdot e(x,t) + s^T(x,t) \cdot e(x,t),
\tag{21}
$$

the induced distribution is identical to the true distribution. In other words, if $e(x,t)$ lives in the null kernel of the operator $\nabla_x + s^T(x,t)$, the trained neural network can perfectly perform the generative task, even if it is not perfectly capturing the score function. We remark that this vector operator is related to the Stein operator (Liu & Wang, 2016) and is the key construct in several recent papers on sampling (Chen & Ghattas, 2020; Fan et al., 2024; Tian et al., 2024). In Fig. 4, we computed the error field on a trained latent diffusion model using forward generated samples, showing that indeed a significant $e(x,t)$ is induced (which is of order $10^2$, significant compared to the order $10^0$ of deterministic decaying flow, $\dot{x}(t) = -x(t)$), but the error field is statistically confined[7] in the null kernel. This analysis suggests that:

> Even when $\text{NN}(x,t)$ is not the score function $\nabla_x \log p(x,t)$, the trained neural network can still be effective to perform generative modeling.

We remark that this analysis is only possible by recognizing the underlying flow structure.

## 5 Discussion and Concluding Remarks

We acknowledge that we are not the first to highlight conceptual link between diffusion and flow-based generative models. Song et al. (2021) recognized the "probability flow" ODE formulation and discussed its use for density and likelihood estimation, while Gao et al. (2025) recently analyzed the formal resemblance between diffusion processes and Gaussian flow matching. In parallel, Ghimire et al. (2023) independently examined the geometric foundations of diffusion models through the lens of Wasserstein geometry, focusing on the Riemannian structure of score-based generative modeling. Their work is complementary to ours but does not explicitly develop the Wasserstein Gradient Flow-Matching interpretation or the connection to the Stein operator emphasized here. To our knowledge, no prior study has unified these perspectives by formulating diffusion modeling as a normalizing flow parameterized through Wasserstein Gradient Flow and Otto calculus. Establishing this link provides a theoretically grounded view that also suggests new avenues for forward random sampling and regularization in flow-based generative learning.

---

[7]We averaged over 256 randomly generated forward samples $x_t$. For each sample, the sufficient condition does not seem to be met but the average seems to agree, noting the significant variance for small $k$.

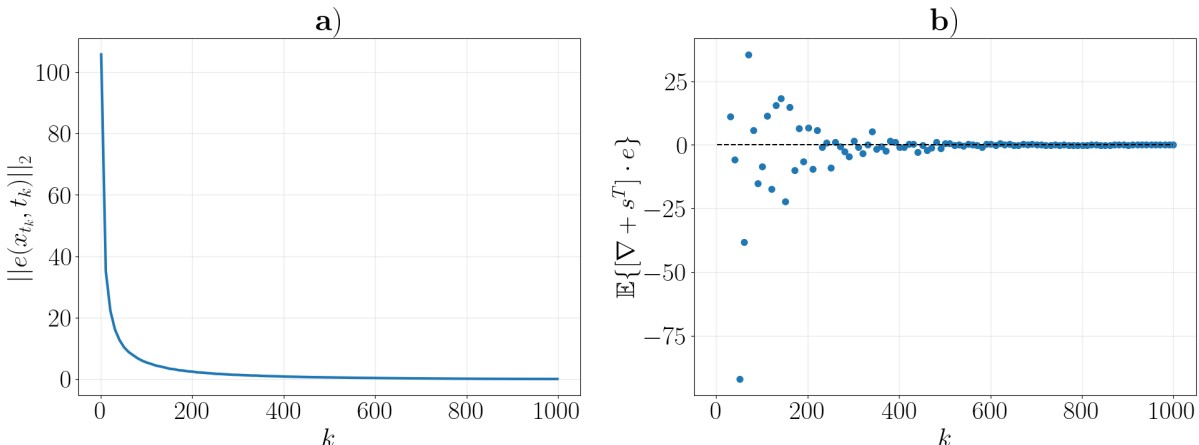

Figure 4: **a)** L2 norm of $e(x, t)$ and **b)** Stein operator value of $e(x, t)$.

To conclude, we advocate for this theoretical framework because first, it was developed over 20 years ago, and yet its direct connection to modern diffusion generative models has received relatively little explicit attention in the machine learning literature; and second, the framework is self-consistent, simple, concise, and elegant. We dedicate this work to the pioneers of Wasserstein Gradient Flow theory—Jordan, Kinderlehrer, and Otto—whose foundational insights continue to shape and inspire cutting-edge machine learning research today.

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

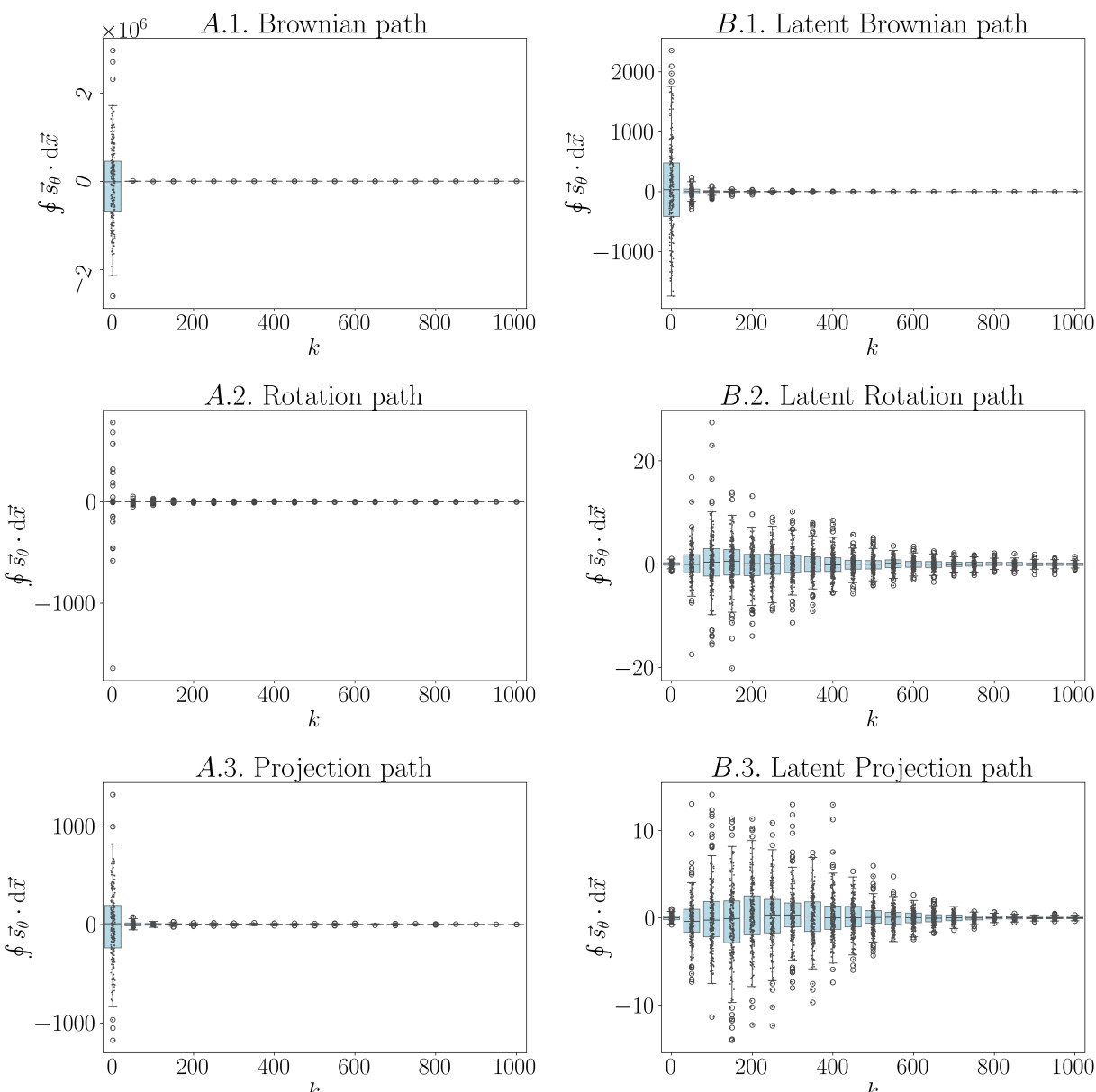

Figure 5: (MNIST) Summary statistics of $\oint \vec{s}_\theta \cdot \mathrm{d}\vec{x}$ calculated by different path-generating mechanisms, in normal and latent diffusions.

# 6 Appendix

We provide more statistics of the non-dimensionalized quantity $|\oint \vec{s}_\theta \mathrm{d}\vec{x}| / \oint |\vec{x}_t| \cdot |\mathrm{d}\vec{x}|$ (Eq. (7)), as well as experiment results on the CIFAR-10 dataset.

## 6.1 Additional numerical results on MNIST

Refer to Figs. 5, 6, 7.

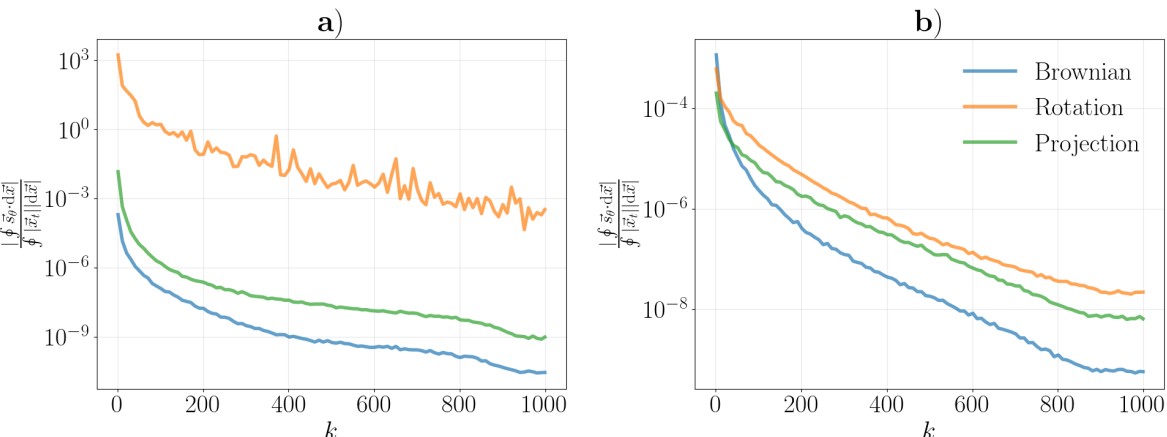

Figure 6: (MNIST) Results of integral constraints, as functions of discrete time index $k$: **a)** shows the absolute value of the integral condition $\oint \vec{s}_\theta \cdot \mathrm{d}\vec{x}$ normalized by the path length and the strength of the deterministic flow, $\oint |\vec{x}_t| |\mathrm{d}\vec{x}|$; **b)** presents the same quantity but for the latent dynamics.

## 6.2 Numerical results on CIFAR-10

For CIFAR-10, we utilized the models from Seong (2024), it implements the standard DDPM and VAE with latent dimension of $3 \times 16 \times 16$. We also tried training these models from scratch, which exhibits similar behaviors to the pretrained ones. Results are presented in Figs. 8, 9, 10, 11.

## 6.3 Additional numerical results on Funnel & CelebA-HQ-256

Summary statistics shown in Fig. 12.

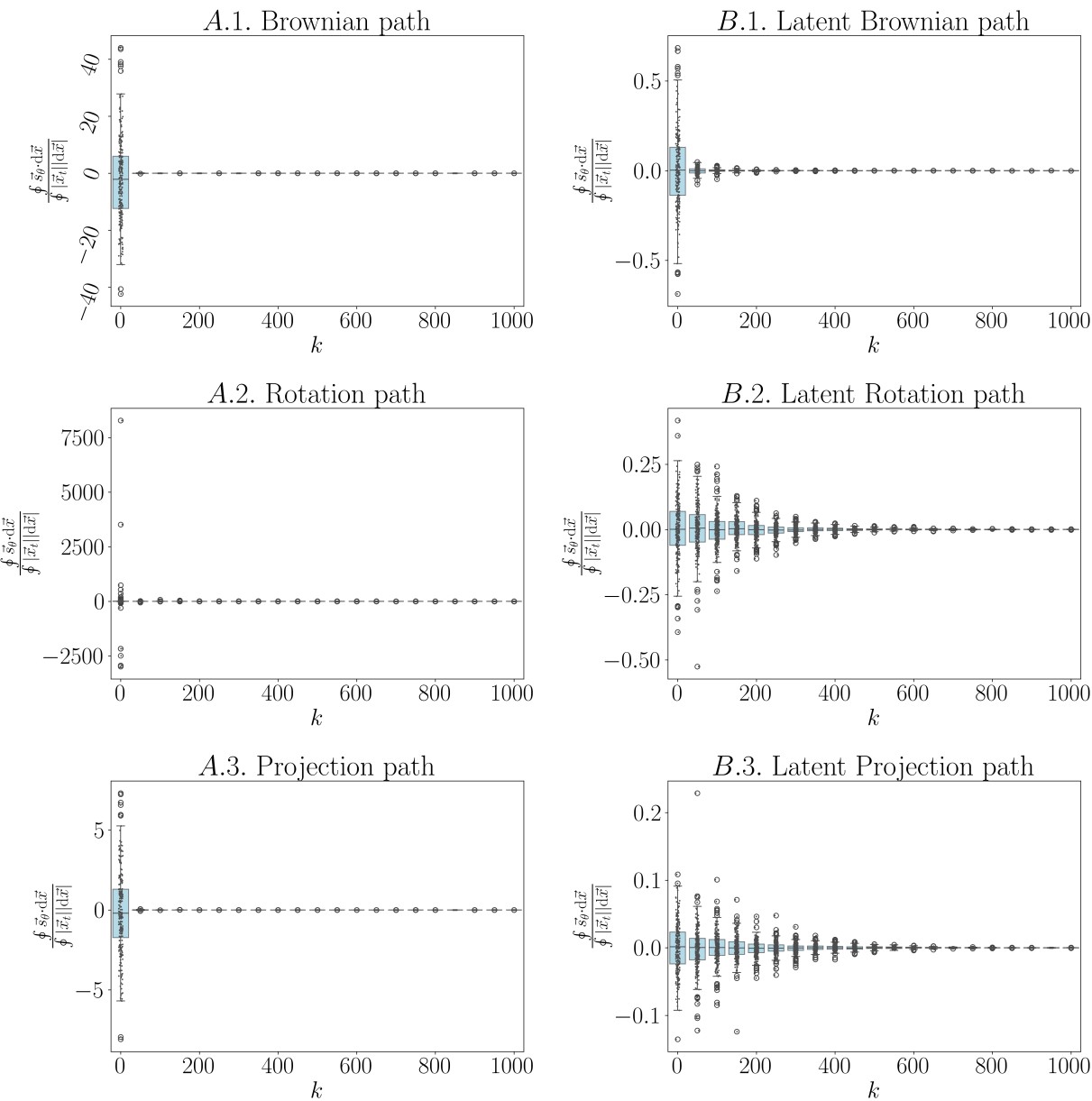

Figure 7: (MNIST) Summary statistics of $|\oint \vec{s}_\theta \cdot d\vec{x}|/\oint |\vec{x}_t||d\vec{x}|$ calculated by different path-generating mechanisms, in normal and latent diffusions.

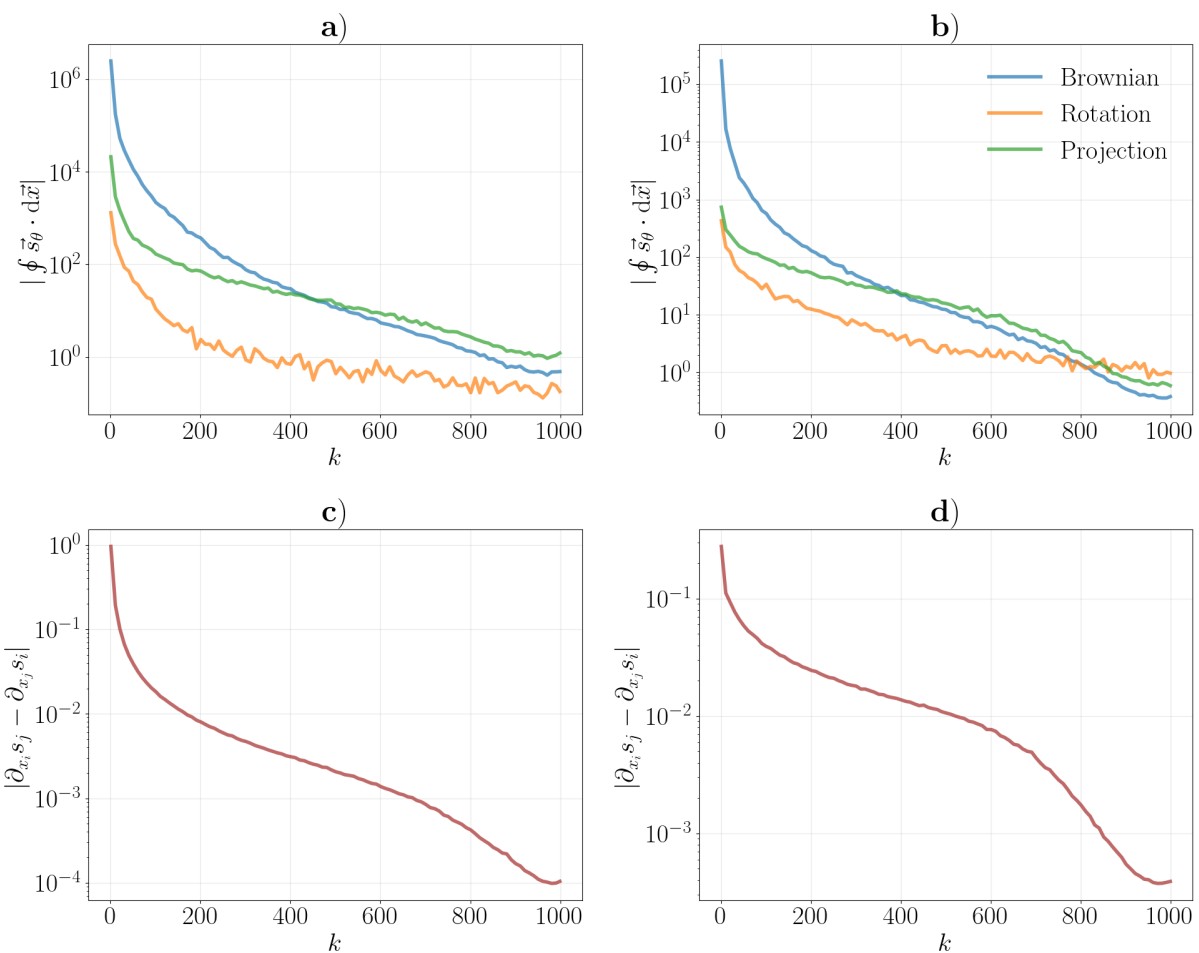

Figure 8: (CIFAR-10) Results of integral and differential constraints, as functions of discrete time index $k$: **a)** shows the absolute value of the integral condition $\oint \vec{s}_\theta \cdot d\vec{x}$; **b)** presents the same quantity but for the latent dynamics; **c)** reports the differential condition $|\partial_{x_i} s_j - \partial_{x_j} s_i|$ in normal diffusion; **d)** shows the corresponding differential condition in latent diffusion.

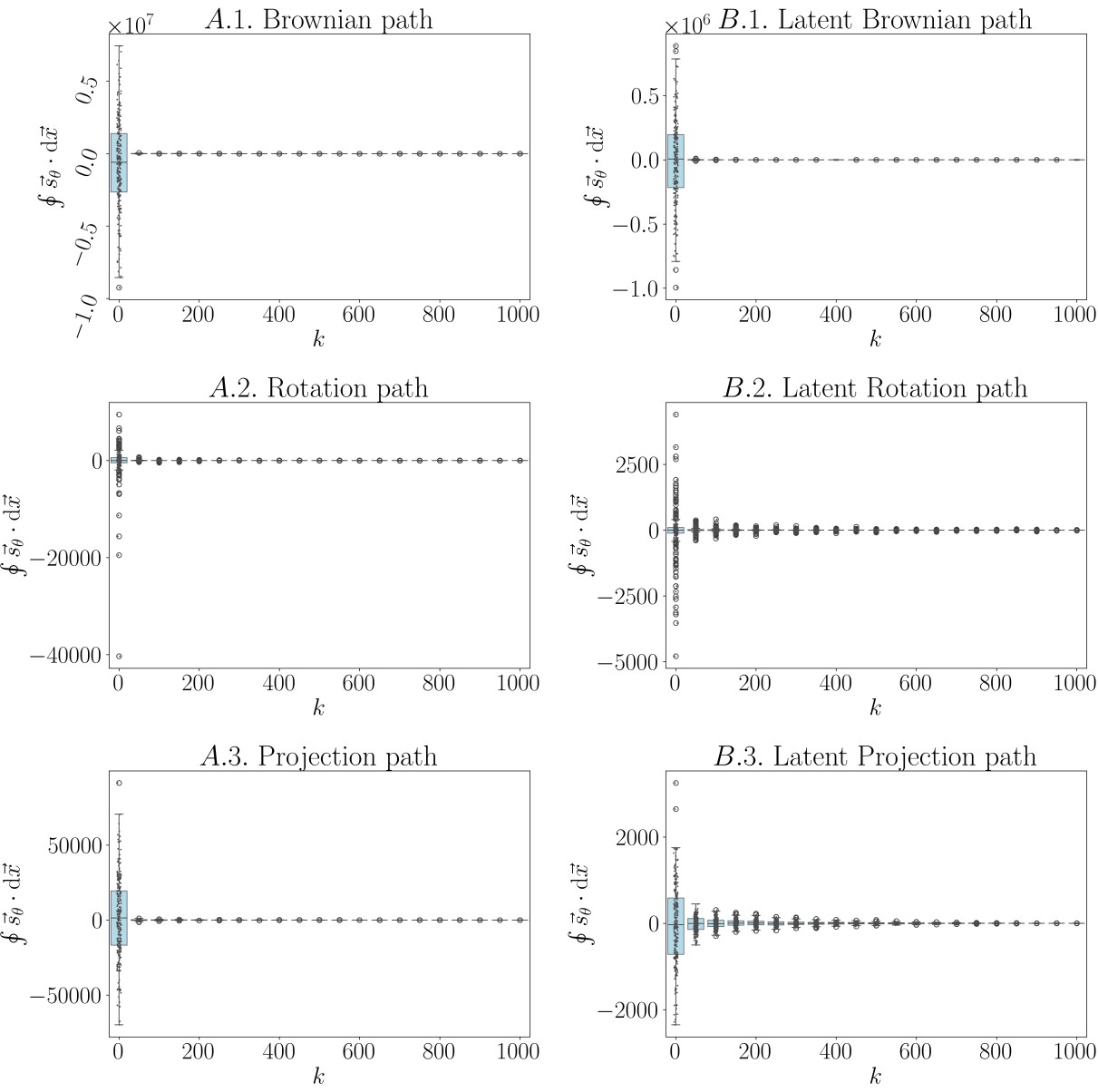

Figure 9: (CIFAR-10) Summary statistics of $\oint \vec{s}_\theta \cdot \mathrm{d}\vec{x}$ calculated by different path-generating mechanisms, in normal and latent diffusions.

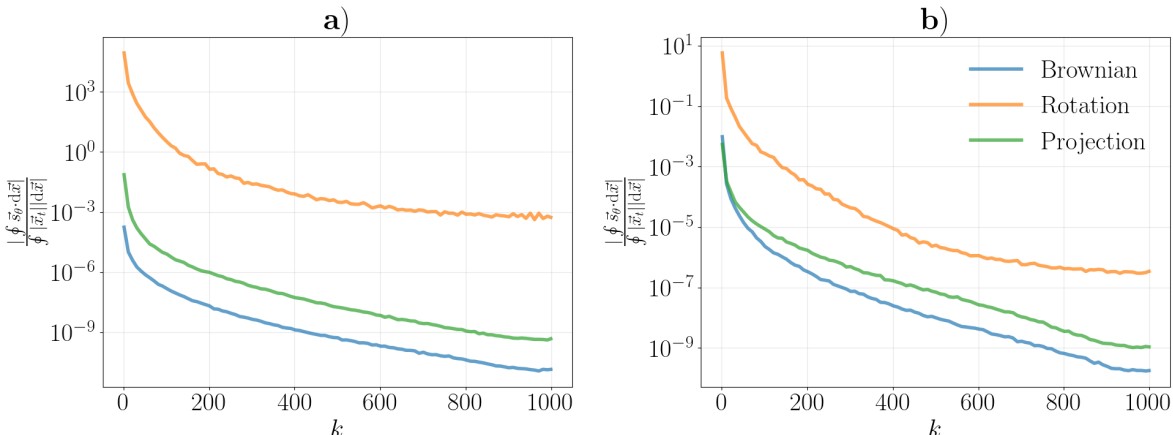

Figure 10: (CIFAR-10) Results of integral constraints, as functions of discrete time index $k$: **a)** shows the absolute value of the integral condition $\oint \vec{s}_\theta \cdot d\vec{x}$ normalized by the path length and the strength of the deterministic flow, $\oint |\vec{x}_t||d\vec{x}|$; **b)** presents the same quantity but for the latent dynamics.

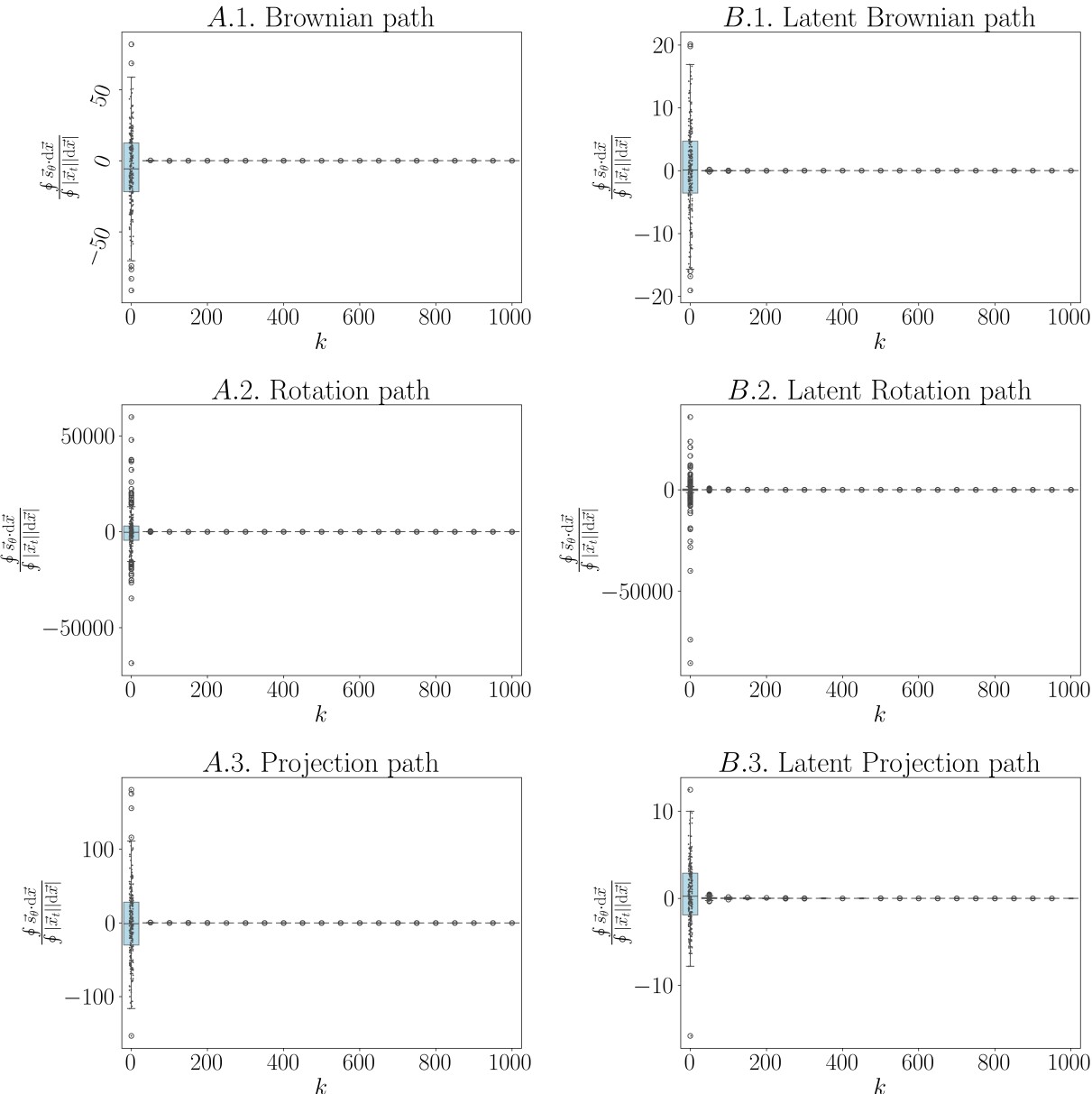

Figure 11: (CIFAR-10) Summary statistics of $|\oint \vec{s}_\theta \cdot d\vec{x}| \oint |\vec{x}_t||d\vec{x}|$ calculated by different path-generating mechanisms, in normal and latent diffusions.

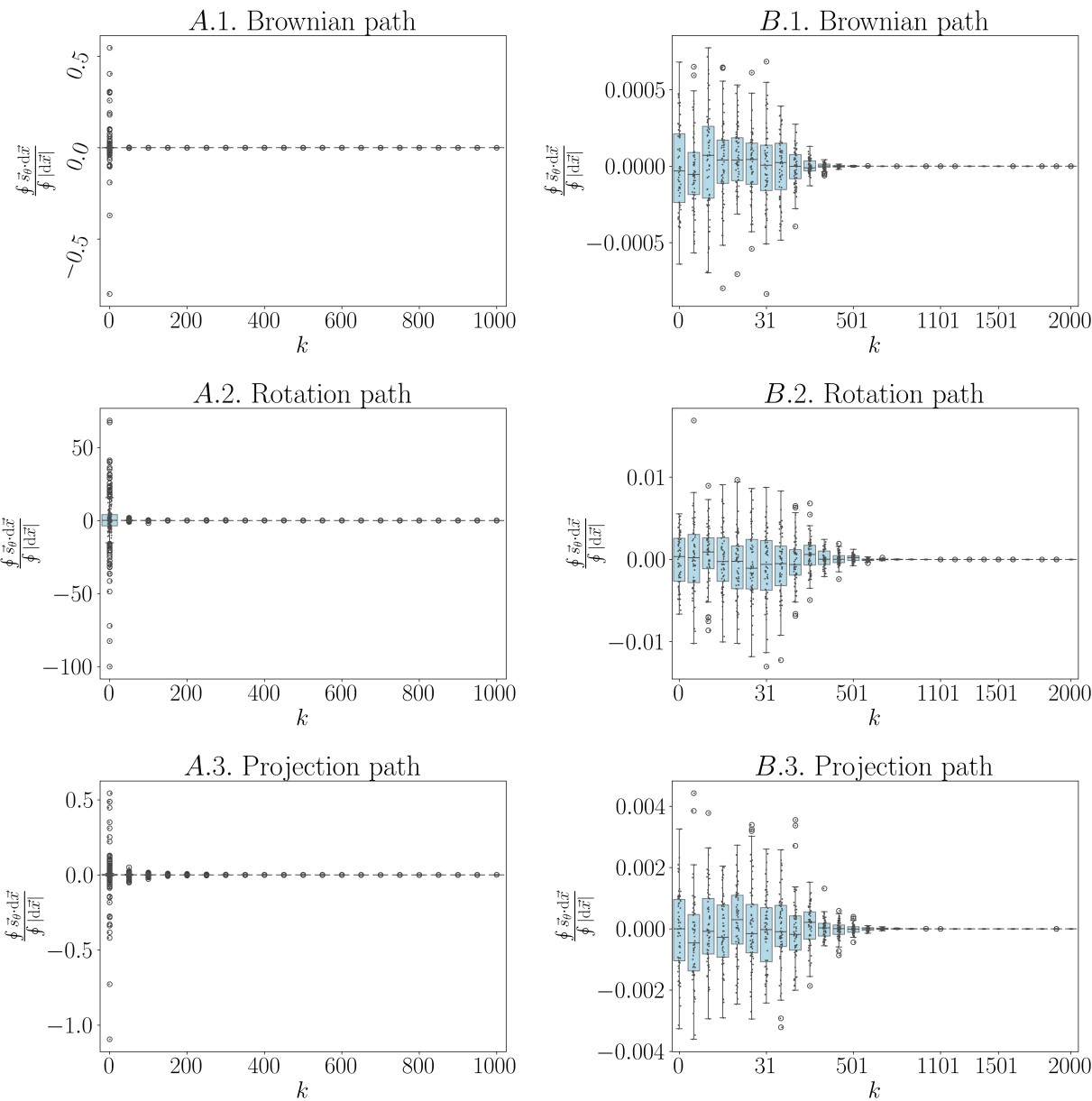

Figure 12: (**Left**: Funnel, **Right**: CelebA-HQ-256) Summary statistics of $|\oint \vec{s}_\theta \cdot d\vec{x}|/\oint |d\vec{x}|$ calculated by different path-generating mechanisms, for data samples drawn from **funnel distribution** (left) and **CelebA-HQ-256** (right). Note that for CelebA-HQ-256, we presented more data points at earlier time $k$.

