# OpenReview forum: "Are We Really Learning the Score Function? Reinterpreting Diffusion Models Through Wasserstein Gradient Flow Matching"
_TMLR — Accepted by TMLR_

### Review · Reviewer_8CrX · 2025-09-25

**Summary Of Contributions:**

This paper offers a reinterpretation of diffusion-based generative models and challenges the conventional perspective that views the neural network as a score function approximator. The authors numerically demonstrated that trained diffusion models do not preserve a conservative vector field, which violates the constraints required of the true score functions. Instead, the authors propose to understand the learned model as performing flow matching to a Wasserstein Gradient Flow. Nonetheless, the authors argue that the generative performance of the diffusion model remains robust because approximate flow matching is sufficient for density transport in practice, even if the learned vector field is not a perfect gradient field.

The key contributions of the paper include: (1) highlighting a mismatch between theory (strict score matching) and practice (non-conservative fields); (2) introducing Wasserstein Gradient Flow as an alternative conceptual perspective on diffusion. A potential limitation is that this paper may overstate the “incorrectness” of standard reverse-time SDE interpretations based on partial or local violations of conservativeness. One could argue that the SDE view remains theoretically valid when given a perfect score matching, while in practice, the score function is approximated by neural networks, and the SDE process is robust to small estimation errors [1]. Nonetheless, the discussion in this paper could deepen connections between diffusion models, optimal transport, and flow-based generative modeling.

[1] Sitan Chen, et al. "Sampling is as easy as learning the score: theory for diffusion models with minimal data assumptions." The Eleventh International Conference on Learning Representations.

**Audience:**

Yes

**Audience Explanation:**

This paper studies the connections between score approximation in diffusion models and Wasserstein gradient flow matching, which is a central topic in the theory of diffusion-based generative models.

**Broader Impact Concerns:**

This paper has no ethical implications.

**Claims And Evidence:**

Yes

**Claims Explanation:**

The theoretical statements are sound. The authors present convincing numerical experiments on various path-based integral evaluations and Jacobian-symmetry checks on MNIST and CIFAR-10. The numerical results support their theoretical claims and demonstrate clear violations of the integral and differential constraints required of gradient fields.

**Requested Changes:**

(1) Discussion on the validity of the SDE frameworks:
As mentioned above, the SDE framework is still valid with a perfect score function, and it is robust to small estimation errors. I suggest that the authors emphasize their theoretical finding as a complementary interpretation of the robustness of diffusion models to imperfect score approximations, and further discuss the effectiveness of the trained neural networks from the perspective of Wasserstein gradient flow matching.

(2) Further evaluation of the integral and differential constraints:
The violation of the integral and differential constraints originates from an imperfect approximation to the true score. Ideally, a neural network with enough capacity is able to fully optimize the objective in equation (15), given sufficient training samples and training steps. Therefore, it is interesting to numerically compare the behaviors of neural networks with different capacities and different training steps and discuss the practical implications of the theoretical findings.

---

> ### Author Response · Authors · 2025-10-20
> **Response to Reviewer 8CrX comments**
>
> > (1) Discussion on the validity of the SDE frameworks: As mentioned above, the SDE framework is still valid with a perfect score function, and it is robust to small estimation errors. I suggest that the authors emphasize their theoretical finding as a complementary interpretation of the robustness of diffusion models to imperfect score approximations, and further discuss the effectiveness of the trained neural networks from the perspective of Wasserstein gradient flow matching.
>
> We thank the reviewer for this constructive suggestion and fully agree with the comment. In our revision, we clarified that our theoretical contribution is not intended to invalidate the reverse-time SDE framework, which remains fully correct when the true score function is known. Rather, our findings complement the SDE view by explaining *why diffusion models trained with neural approximations can still perform robustly, even when the learned vector field deviates from a true conservative score.*
>
> In particular, we emphasize in the revised manuscript that the Wasserstein Gradient Flow (WGF) interpretation offers an alternative but compatible perspective: it accounts for the empirical observation that imperfect “score” learning still yields correct marginal density transport. From this viewpoint, the effectiveness of diffusion models arises from performing approximate flow matching to the WGF velocity field, which remains sufficient to preserve the desired density evolution.
>
> To make this explicit, we revised the relevant section (formerly Section 3, now Section 4 in the updated draft) to read:
> *“The WGF view complements rather than replaces the reverse-SDE interpretation. When the score function is exact, both formulations describe the same density dynamics; when the score is approximate, the WGF formulation naturally explains the empirical robustness of diffusion models as approximate flow matching to the correct Wasserstein gradient field.”*
>
> >(2) Further evaluation of the integral and differential constraints: The violation of the integral and differential constraints originates from an imperfect approximation to the true score. Ideally, a neural network with enough capacity is able to fully optimize the objective in equation (15), given sufficient training samples and training steps. Therefore, it is interesting to numerically compare the behaviors of neural networks with different capacities and different training steps and discuss the practical implications of the theoretical findings.
>
> We thank the reviewer for this suggestion. Following this recommendation, we have extended our analysis to include larger models trained on bigger datasets, including the pretrained checkpoints provided by Song et al. (2021). In the added datasets/models/problem setups (please also see Response 1.2), we observed consistent violations of the integral and differential constraints. These results confirm that the non-conservative behavior is not simply a result of limited model capacity or undertraining, but a more general feature of current diffusion training practices.

---

### Review · Reviewer_w6So · 2025-09-28

**Summary Of Contributions:**

This work analyses diffusion models. It first provides numerical evidences that the usual practice of approximating the score function with a neural network, does not actually learn the score function as the neural networks are not constrained to produce conservative fields. More preciselly, there are two constraints satisfied by scores: the integral along closed path should be null, and a differential constraint. And the authors show that they are usually not respected. Then, they raise the question how it is still able to perform generative modeling, and propose to see it through the lens of flow matching, where the velocity field would be chosen as the Wasserstein gradient of the KL with a standard Gaussian as target, as its Wasserstein gradient flow corresponds to the forward SDE solved in diffusion models.

**Additional Comments:**

I think there are some interesting things in this work such as the remarks on the score which is not well learned when doing diffusion models, and the proposition to use instead a Flow-Matching loss to learn the velocity field instead.

However, the paper could be improved a lot on the organization and writing, the connections with related works, and the analysis of the proposed algorithm. In particular, I think it should be compared with score-based models, flow matchings, and on image data.

**Audience:**

Yes

**Audience Explanation:**

Understanding better diffusion models is an important field, and thus this paper could interest a large audience of TMLR.

**Claims And Evidence:**

Yes

**Claims Explanation:**

The claims made in the submission are mostly accurate. The numerical evidence that neural networks used to approximate the score actually don't learn the score seem accurate. And the proposition to see this problem through the lens of flow matching is interesting.

However, there are few things that could be improved, few statements which does not seem totally right, and the connection to some relevant relative works is missing. Moreover, the proposed algorithm using Flow Matching is not shown to work, and not very clear.

**Requested Changes:**

First, I believe that the paper could be better written and organized. For instance, in the Background, there is a Section 1.1, but not Section 1.2; there is no conclusion; the relation with flow matching, which is an important part of the paper is in a Section "Discussion". I would advice to improve the presentation of the paper.

In Section 1.1, it is said that "each timestep corresponds to minimizing a free energy functional that combines Shannon entropy with a Wasserstein-2 distance penalty". But for the equivalence between the Fokker-Planck equation, and the Wasserstein gradient flow, the minimized functional is the relative entropy (Kullback-Leibler divergence), not just the entropy that would give the Heat PDE.

In Section 3, it is not clear if the Wasserstein Gradient Flow-Matching is actually used in practice. I would expect it to be compared with the usual score-based diffusion method, to verify that it works as well or not.

Moreover, I did not understand some things about the algorithm. It is stated page 6 "More precisely, we only match the flow induced by the entropic term in equation 4.". Does it change anything to learn only the entropic part of the flow, and not the full velocity field? In the Data Generation step, it is said that $s(x,t)$ is computed analytically. Could you precise how it is computed? Is there a closed-form? (I know there is one at least if $\rho_0$ is Gaussian but is it the case more generally?)

When I see this algorithms, I can't help of thinking that it will be much slower than usual flow matchings since you need first to do the forward step to get all the time steps. How does it compare?

The fact that the backward process is equivalent to a Wasserstein Gradient Flow of the functional given in equation (18) has already been observed in [1] (Theorem 2). Moreover in this paper, they also observed the relations between Wasserstein gradient flows and diffusion. A discussion of this paper, and what is different should be added.

The claim page 8 that this theoretical framework "has been largely ignored in the machine learning literature" is wrong, as there is a big literature on Wasserstein gradient flows to solve many tasks ranging from sampling to generative modeling, see e.g. not exhaustively [2,3,4,5] for sampling and [6,7,8,9,10] for generative modeling.


Typos:
- Equation (13): lacks "."
- Equation (18): "$\rho(x,s)$" in first line


[1] Ghimire, S., Liu, J., Comas, A., Hill, D., Masoomi, A., Camps, O., & Dy, J. (2023). Geometry of score based generative models. arXiv preprint arXiv:2302.04411.

[2] Wibisono, A. (2018, July). Sampling as optimization in the space of measures: The Langevin dynamics as a composite optimization problem. In Conference on learning theory (pp. 2093-3027). PMLR.

[3] Salim, A., Korba, A., & Luise, G. (2020). The Wasserstein proximal gradient algorithm. Advances in Neural Information Processing Systems, 33, 12356-12366.

[4] Lambert, M., Chewi, S., Bach, F., Bonnabel, S., & Rigollet, P. (2022). Variational inference via Wasserstein gradient flows. Advances in Neural Information Processing Systems, 35, 14434-14447.

[5] Huix, T., Korba, A., Durmus, A., & Moulines, E. (2024). Theoretical guarantees for variational inference with fixed-variance mixture of Gaussians. arXiv preprint arXiv:2406.04012.

[6] Alvarez-Melis, D., Schiff, Y., & Mroueh, Y. (2021). Optimizing functionals on the space of probabilities with input convex neural networks. arXiv preprint arXiv:2106.00774.

[7] Mokrov, P., Korotin, A., Li, L., Genevay, A., Solomon, J. M., & Burnaev, E. (2021). Large-scale wasserstein gradient flows. Advances in Neural Information Processing Systems, 34, 15243-15256.

[8] Fan, J., Zhang, Q., Taghvaei, A., & Chen, Y. (2021). Variational Wasserstein gradient flow. arXiv preprint arXiv:2112.02424.

[9] Choi, J., Choi, J., & Kang, M. (2024). Scalable Wasserstein gradient flow for generative modeling through unbalanced optimal transport. arXiv preprint arXiv:2402.05443.

[10] Terpin, A., Lanzetti, N., Gadea, M., & Dörfler, F. (2024). Learning diffusion at lightspeed. Advances in Neural Information Processing Systems, 37, 6797-6832.

---

> ### Author Response · Authors · 2025-10-20
> **Response to Reviewer w6So comments, part 1**
>
> > First, I believe that the paper could be better written and organized. For instance, in the Background, there is a Section 1.1, but not Section 1.2; there is no conclusion; the relation with flow matching, which is an important part of the paper is in a Section "Discussion". I would advice to improve the presentation of the paper.
>
> We thank the reviewer for the helpful feedback. We have substantially improved the organization and clarity of the paper by adding a proper introduction, a dedicated related-work section covering both recent and earlier studies, and an enhanced discussion. The missing “1.2” subsection was a formatting artifact in the template, which has now been corrected. Overall, the presentation has been thoroughly revised and strengthened.
>
> > In Section 1.1, it is said that "each timestep corresponds to minimizing a free energy functional that combines Shannon entropy with a Wasserstein-2 distance penalty". But for the equivalence between the Fokker-Planck equation, and the Wasserstein gradient flow, the minimized functional is the relative entropy (Kullback-Leibler divergence), not just the entropy that would give the Heat PDE.
>
> We thank the reviewer for pointing out this important clarification. We have revised the text accordingly to avoid potential confusion. In the updated manuscript, we now state that “each timestep corresponds to minimizing a free-energy functional composed of a potential energy term $V(x)$ and a negative Shannon entropy of the evolving distribution, plus a Wasserstein-2 distance term between successive densities (scaled by the time step). The entropy and the Wasserstein-2 distance can both be viewed as regularizers of the optimization problem for the potential energy. Moreover, the sum of the potential energy and the negative entropy is equivalent to the Kullback-Leibler divergence $\text{KL}(\rho_t \Vert \pi)$ from the evolving probability density $\rho_t$ to the Gibbs measure $\pi(x) \propto \exp(-V(x))$.”
>
> > In Section 3, it is not clear if the Wasserstein Gradient Flow-Matching is actually used in practice. I would expect it to be compared with the usual score-based diffusion method, to verify that it works as well or not.
> >
> > Moreover, I did not understand some things about the algorithm. It is stated page 6 "More precisely, we only match the flow induced by the entropic term in equation 4.". Does it change anything to learn only the entropic part of the flow, and not the full velocity field? In the Data Generation step, it is said that is computed analytically. Could you precise how it is computed? Is there a closed-form? (I know there is one at least if is Gaussian but is it the case more generally?)
> >
> > When I see this algorithms, I can't help of thinking that it will be much slower than usual flow matchings since you need first to do the forward step to get all the time steps. How does it compare?
>
> We thank the reviewer for raising this point. We clarify that, operationally, the learning and inference procedures in our work are identical to those used in standard score-based diffusion models. As noted in the submitted manuscript:
>
> *“Operationally, the above descriptions are identical to applying the ‘score-matching’ for training and performing ‘probability flow’ for inference (Song et al.).”*
>
> Our contribution is not to introduce a new algorithmic variant, but to provide a theoretical reinterpretation of existing diffusion methods through the lens of Wasserstein Gradient Flow (WGF). The WGF formulation clarifies *why the conventional score-matching and probability-flow algorithms succeed even when the learned vector field is not strictly conservative.*
>
> Regarding the phrase “we only match the flow induced by the entropic term,” we meant that in our practical implementation, the learnable neural velocity field corresponds to the entropy-induced component of the WGF velocity $-\nabla_x \log \rho(x,t)$, since the drift term (the potential component $-x$) is analytically known. Thus, this is simply a choice of parametrization that does not omit any learnable dynamics: it merely separates the analytically tractable part from the learned part. We remark that this is also the choice in existing VP algorithms, such as DDPM or VP for SDE. For Gaussian forward processes, this decomposition indeed has a closed form, due to the analytical O-U process; for more general data distributions, we rely on the same Monte Carlo sampling strategy used in conventional diffusion training to evaluate the score target at each noise level.
>
> Finally, the computational cost is the same as standard diffusion models. In practice this corresponds exactly to the usual noising process already performed in standard DDPM or SDE-based training. Therefore, no additional forward passes or time complexity are introduced.

---

> ### Author Response · Authors · 2025-10-20
> **Response to Reviewer w6So comments, part 2**
>
> > The fact that the backward process is equivalent to a Wasserstein Gradient Flow of the functional given in equation (18) has already been observed in [1] (Theorem 2). Moreover in this paper, they also observed the relations between Wasserstein gradient flows and diffusion. A discussion of this paper, and what is different should be added.
>
> We agreed. In fact, we acknowledge that this was known as the probability flow. We thank the reviewer for bringing up this preprint, which we found has not been published yet and thus is not in our literature pool. We tried to integrate the work and have performed a major revision of the manuscript, hoping to highlight the points. The revision includes in the (newly added, stand-alone) literature review, *“[1] investigated the Riemannian structure underlying score-based generative models, providing valuable insights into their manifold properties. However, their goal is geometric analysis of existing models, not reinterpretation of the algorithm through our proposed WGF flow matching.”* and in the discussion, *“In parallel, [1] independently examined the geometric foundations of diffusion models through the lens of Wasserstein geometry, focusing on the Riemannian structure of score-based generative modeling. Their work is complementary to ours but does not explicitly develop the Wasserstein flow-matching interpretation or the connection to the Stein operator emphasized here.”*
>
> [1] Ghimire, S., Liu, J., Comas, A., Hill, D., Masoomi, A., Camps, O., & Dy, J. (2023). Geometry of score based generative models. arXiv preprint arXiv:2302.04411.
>
> > The claim page 8 that this theoretical framework "has been largely ignored in the machine learning literature" is wrong, as there is a big literature on Wasserstein gradient flows to solve many tasks ranging from sampling to generative modeling, see e.g. not exhaustively [2,3,4,5] for sampling and [6,7,8,9,10] for generative modeling.
>
> We thank the reviewer for pointing out this important clarification. We agree that there is now a substantial and rapidly growing body of work employing Wasserstein Gradient Flow (WGF) formulations for both sampling and generative modeling tasks (e.g., [2–10]). Our original statement was not meant to overlook these important contributions, but rather to emphasize that the specific connection between diffusion models, the JKO/Otto calculus framework, and the flow-matching interpretation has remained comparatively underexplored in the modern machine learning literature.
>
> To avoid misunderstanding, we have revised the relevant sentence to read:
>
> “To conclude, we advocate for this theoretical framework because, first, it was developed over 20 years ago and yet its direct connection to modern diffusion generative models has received relatively little explicit attention in the machine learning literature…”
>
> > Typos:
> >
> >  Equation (13): lacks "."
> >
> > Equation (18):  “$rho(x,s)$" in first line
>
> We thank the reviewer for identifying these errors. We have corrected both instances.

---

> > ### Comment · Reviewer_w6So · 2025-10-22
> >
> > Thank you for your response and taking into account my review. I find the paper much clearer now.

---

### Review · Reviewer_sdU7 · 2025-10-06

**Summary Of Contributions:**

The paper revisits the theoretical interpretation of diffusion generative models.
Traditionally, diffusion models are trained to approximate the score function (the gradient of the log-density of the noisy data) and use it to define a reverse-time SDE for sampling.
The authors argue that, in practice, trained networks do not produce conservative vector fields (a requirement for being true scores) and thus may not actually learn the score function.
They provide numerical evidence by checking integral and differential conditions that a true score must satisfy and find them violated for trained diffusion models (on MNIST and a latent-diffusion variant).
To reconcile this, the paper proposes viewing diffusion models as performing flow matching to the velocity field of a Wasserstein Gradient Flow (WGF) rather than strict score learning.
They further connect the resulting error field to the Stein operator, showing why models can still transport densities correctly even when the learned field is not a true score.

**Audience:**

Yes

**Audience Explanation:**

Yes. The reinterpretation of diffusion models as performing Wasserstein Gradient Flow (WGF) flow matching rather than strict score learning is timely and relevant to the machine learning theory and generative modeling communities. Many researchers are currently interested in the foundations of diffusion models and in connecting them to optimal transport and normalizing flows. Even if the empirical validation is limited, the conceptual reframing could influence how future algorithms are designed and analyzed. Readers who work on the mathematical underpinnings of generative models, as well as those exploring new training objectives beyond score matching, would likely find the ideas and the error analysis around Eq. (19–20) valuable.

**Broader Impact Concerns:**

Nothing to remark

**Claims And Evidence:**

No

**Claims Explanation:**

The paper presents a well-motivated reinterpretation of diffusion models as performing Wasserstein Gradient Flow (WGF) flow matching rather than strictly learning the score function of the noisy data. The derivations are clear and appear mathematically sound, and the empirical checks on integral and differential constraints (Eqs. 11–12) provide an interesting first indication that trained diffusion networks may not produce conservative vector fields.

That said, the empirical support for the strongest claim — that diffusion models do not in practice learn the true score function — feels limited. The numerical analysis is carried out only on a single lightweight U-Net trained on MNIST and a latent-diffusion variant. While these experiments show that the integral and differential constraints are violated for those models, this does not convincingly demonstrate that more expressive architectures, larger datasets, or more thorough optimization could not approximate a true score. In the limit of highly flexible models and sufficient training, one might still expect the score function to be reachable.

In addition, the paper introduces an interesting characterization of the approximation error  $e(x,t)$ through Eq. (19–20) and its relation to the Stein operator. However, this insight is not validated in a controlled synthetic setting where the true score and its error are known — for example, in low-dimensional Gaussian mixtures or other analytically tractable distributions. Such experiments could test whether diffusion models fundamentally fail to learn the score even when the target density is simple and fully known.

Overall, the evidence is thought-provoking and suggests that current training practices do not enforce conservativeness, but it is not yet strong enough to make the broader claim that diffusion models cannot or generally do not learn the score function. Strengthening the empirical validation — for instance, through synthetic experiments and larger models — would make the central claims more convincing.

**Requested Changes:**

**Critical changes (needed to support acceptance)**:

- Strengthen empirical validation of the main claim.

Test whether the violation of the integral and differential constraints (Eqs. 11–12) persists with larger and more expressive models, different datasets, or more thorough optimization.

Include at least one controlled synthetic experiment where the true score is analytically known (e.g., Gaussian mixtures or other low-dimensional tractable densities). This would allow the authors to verify whether the observed violations are fundamental or specific to the tested lightweight models.

- Clarify the scope of the claim.

Specify whether the work argues that current training practice does not guarantee learning the true score, or whether it suggests that learning the true score is fundamentally unnecessary or unlikely even for very flexible models.

Make it clear whether the WGF view is meant to fully replace the SDE/score-based view or to serve as an alternative but compatible perspective.

**Important but not critical (would strengthen the paper):**


- Offer a short conceptual explanation or visual intuition for Wasserstein Gradient Flow and Otto calculus for readers without deep background in optimal transport.

- Clearly separate theoretical reformulation from empirical observations to help non-experts follow the argument.

- Explore practical implications. Discuss how imperfect score learning might affect generative quality, density estimation, or mode coverage (similar to issues known in GANs).

---

> ### Author Response · Authors · 2025-10-20
> **Response to Reviewer sdU7 comments, part 1**
>
> We thank the review for the thoughtful and constructive comment. We appreciate the reviewer’s clear summary of our work and for recognizing the motivation and clarity of our theoretical framework. We are glad that the reinterpretation of diffusion models as performing Wasserstein Gradient Flow (WGF) flow matching rather than strict score learning came through clearly, and that the empirical validation of the integral and differential constraints was found informative.
>
> > this does not convincingly demonstrate that more expressive architectures, larger datasets, or more thorough optimization could not approximate a true score
>
> >  for example, in low-dimensional Gaussian mixtures or other analytically tractable distributions. Such experiments could test whether diffusion models fundamentally fail to learn the score even when the target density is simple and fully known.
>
> > Test whether the violation of the integral and differential constraints (Eqs. 11–12) persists with larger and more expressive models, different datasets, or more thorough optimization.
> >
> > Include at least one controlled synthetic experiment where the true score is analytically known (e.g., Gaussian mixtures or other low-dimensional tractable densities). This would allow the authors to verify whether the observed violations are fundamental or specific to the tested lightweight models.
>
> We thank the reviewer for this suggestion. Following their recommendation, we conducted additional experiments to strengthen the empirical support of our claims. Specifically, we evaluated the integral and differential constraints on (1) CIFAR-10 dataset with latent DDPM, (2) Variance Exploding SDE model on CelebA-HQ-256 released by Song et al. (2021), and (3) the analytical funnel distribution proposed by Neal.
>
> We observed similar behaviors of the score field reported in the paper, confirming that these effects persist even in (a) simple distributions, (b) large-scale models, (c) larger datasets, and (d) well-optimized diffusion architectures.
>
> These new results demonstrate that the non-conservativeness we highlight is not an artifact of lightweight architectures or limited training, but a more universal property of diffusion model learning dynamics.
>
> Regarding the suggestion of a controlled synthetic experiment with a known analytical score function, we appreciate the reviewer’s intent. However, even for seemingly simple non-Gaussian distributions—such as the funnel distribution—the closed-form expression of the time-evolved score under Ornstein–Uhlenbeck corruption is analytically intractable for arbitrary $t>0$. For mixtures of $N$ Gaussians (we remark that, for any finite dataset, existing diffusion modeling approach is already generating $N$-mode Gaussian mixture), the exact score is can be derived but scales as $\mathcal{O}(N)$, which quickly becomes computationally prohibitive except for toy cases. Conceptually, since the true score field by definition satisfies both the integral and differential constraints, demonstrating systematic violations of these constraints by trained diffusion networks already establishes that the learned field deviates from the true score.
> We hope these additional results and clarifications address the reviewer’s concern. We would welcome further specific suggestions if additional synthetic scenarios are of particular interest.

---

> ### Author Response · Authors · 2025-10-20
> **Response to Reviewer sdU7 comments, part 2**
>
> > Specify whether the work argues that current training practice does not guarantee learning the true score, or whether it suggests that learning the true score is fundamentally unnecessary or unlikely even for very flexible models.
> >
> > Make it clear whether the WGF view is meant to fully replace the SDE/score-based view or to 3serve as an alternative but compatible perspective.
>
> > - Offer a short conceptual explanation or visual intuition for Wasserstein Gradient Flow and Otto calculus for readers without deep background in optimal transport.
> > - Clearly separate theoretical reformulation from empirical observations to help non-experts follow the argument.
> > - Explore practical implications. Discuss how imperfect score learning might affect generative quality, density estimation, or mode coverage (similar to issues known in GANs).
>
> Our results indicate that while diffusion models do not learn the *true* score function, they still perform the intended generative task effectively. In the submitted version, we had made it clear in the abstract: “We present numerical evidence that trained diffusion networks violate both integral and differential constraints required of true score functions, demonstrating that the learned vector fields are not conservative. Despite this, the models perform remarkably well as generative mechanisms.” Therefore, our work argues that learning the exact score is not required for enabling a dynamical system that transports the probability distribution for sample generation. Furthermore, we show that Wasserstein Gradient Flow can be this dynamical system, and it can be learned through flow matching, which surprisingly results in the same learning and inference procedures. In summary, our contributions include (1) demonstrating the violation of integral and differential constraints, showing that the true score function is not learned, and (2) establishing the theoretical interpretation of why existing procedures work well for generative modeling despite the violations through the WGF theory. In this revision, we have performed a major revision and restructuring of the paper, with the hope that the restructuring will make our chain of logic clearer.

---

> > ### Comment · Reviewer_sdU7 · 2025-10-28
> >
> > Thank you very much for all your clarifications. I believe they have significantly improved the paper and strengthened the empirical validation of the proposed ideas.

---

### Decision · Action_Editor_TNVq · 2025-11-06

**Recommendation:** Accept as is

**Audience:**

Yes

**Audience Explanation:**

The reviewers agreed that the paper's topic is closely related to the TMLR community.

**Claims And Evidence:**

Yes

**Claims Explanation:**

This paper provides a novel understanding of how imperfect score estimation that does not satisfy integral and differential constraints can still generate good samples in practice. The reviewers found that all claims are sound and supported by both theoretical explanations and numerical validations.